# Medical Grade Honey as a Promising Treatment to Improve Ovarian Tissue Transplantation

**DOI:** 10.3390/bioengineering9080357

**Published:** 2022-07-30

**Authors:** Ana Rita Azevedo, Ana Sofia Pais, Teresa Almeida-Santos, Virgínia M. R. Pires, Pedro Pessa, Carla C. Marques, Sofia Nolasco, Pedro Castelo-Branco, José A. M. Prates, Luís Lopes-da-Costa, Mafalda Laranjo, Maria Filomena Botelho, Rosa M. L. N. Pereira, Jorge M. B. G. A. Pimenta

**Affiliations:** 1INIAV—Instituto Nacional de Investigação Agrária e Veterinária I.P., Unidade de Biotecnologia e Recursos Genéticos, Quinta da Fonte Boa, 2005-048 Vale de Santarém, Portugal; nvarandamarques@gmail.com (C.C.M.); rosa.linoneto@iniav.pt (R.M.L.N.P.); jorgepimenta7@gmail.com (J.M.B.G.A.P.); 2Reproductive Medicine Unit, Centro Hospitalar e Universitário de Coimbra, 3000-075 Coimbra, Portugal; asfpais@uc.pt (A.S.P.); anateresasantos.tas@gmail.com (T.A.-S.); 3Faculty of Medicine, University of Coimbra, Azinhaga de Santa Comba, Celas, 3000-548 Coimbra, Portugal; 4Institute of Biophysics, Faculty of Medicine, University of Coimbra, 3000-548 Coimbra, Portugal; mafaldalaranjo@gmail.com (M.L.); mfbotelho@fmed.uc.pt (M.F.B.); 5Coimbra Institute for Clinical and Biomedical Research (iCBR), Faculty of Medicine, University of Coimbra, 3000-548 Coimbra, Portugal; 6Center for Innovative Biomedicine and Biotechnology (CIBB), University of Coimbra, 3000-548 Coimbra, Portugal; 7Centre of Investigation in Environment, Genetics and Oncobiology (CIMAGO), Faculty of Medicine, University of Coimbra, 3000-548 Coimbra, Portugal; 8Clinical Academic Center of Coimbra (CACC), 3004-561 Coimbra, Portugal; 9CNC—Center for Neuroscience and Cell Biology, CIBB, University of Coimbra, Azinhaga de Santa Comba, Celas, 3004-504 Coimbra, Portugal; 10CIISA—Centre for Interdisciplinary Research in Animal Health, Faculdade de Medicina Veterinária, Universidade de Lisboa, 1300-477 Lisboa, Portugal; virginia.pires@nzytech.com (V.M.R.P.); sofia.nolasco@gmail.com (S.N.); japrates@fmv.ulisboa.pt (J.A.M.P.); lcosta@fmv.ulisboa.pt (L.L.-d.-C.); 11NZYTech—Genes and Enzymes, Campos do Lumiar, Edifício E, 1649-038 Lisboa, Portugal; 12Serviço de Anatomia Patológica, Centro Hospitalar e Universitário de Coimbra, 3000-075 Coimbra, Portugal; p.pessa59@gmail.com; 13ESTeSl—Escola Superior de Tecnologia da Saúde de Lisboa, Instituto Politécnico de Lisboa, 1990-096 Lisboa, Portugal; 14Faculdade de Medicina, Universidade do Algarve, 8005-139 Faro, Portugal; castelobranco.pedro@gmail.com; 15Laboratório Associado para Ciência Animal e Veterinária (AL4AnimalS), Faculdade de Medicina Veterinária, Universidade de Lisboa, 1300-477 Lisboa, Portugal

**Keywords:** xenografting, ovarian, vitrification, honey, fertility, cryopreservation

## Abstract

Ovarian tissue cryopreservation is a female fertility preservation technique that presents major challenges for the maintenance of follicular viability after transplantation. The aim of this study was to evaluate and compare the application of L-Mesitran Soft^®^, a product containing 40% medical grade honey (MGH), with other strategies to improve ovarian grafts’ viability. For this purpose, bovine ovarian tissue was vitrified, warmed and randomly assigned to culture groups: (1) control, (2) MGH 0.2% in vitro, (3) MGH in vivo (direct application in the xenotransplantation), (4) vascular endothelial growth factor (VEGF 50 ng/mL) and (5) vitamin D (100 Nm), during a 48 h period. A sixth group (6) of fragments was thawed on transplantation day and was not cultured. The tissue was xenotransplanted into immunodeficient (Rowett nude homozygous) ovariectomized rats. Grafts were analyzed 48 h after culture, and 7 and 28 days after transplantation. The tissue was subjected to histological and immunohistochemical analysis. Treatments using MGH showed the highest angiogenic and cell proliferation stimulation, with cellular apoptosis, within a healthy cellular turnover pathway. In conclusion, MGH should be considered as a potentially effective and less expensive strategy to improve ovarian tissue transplantation.

## 1. Introduction

Fertility preservation strategies have emerged with the evolution of assisted reproductive technologies [1,2]. Ovarian tissue cryopreservation is the main option for prepubertal females and for those who cannot delay cancer treatment. This technique offers great potential, as it can preserve thousands of ovarian follicles at one time, allowing restoration not only of fertility, but also of endocrine function [3,4].

One crucial factor for the success of the ovarian tissue transplantation technique is directly related to the speed of revascularization, due to ischemic injuries. The ischemia that occurs until the establishment of graft revascularization leads to the loss of more than 50% of primordial follicles. This occurs predominantly in the first 5 days after transplantation and can compromise the durability of the transplant [4,5,6,7]. Thus, angiogenesis is important, since hypoxia and ischemia are two critical factors for post-transplant survival.

Angiogenesis is a complex physiological process regulated by multiple angiogenic factors, components of the extracellular matrix and endothelial cells [8]. In women, angiogenesis is particularly important for the function of the reproductive organs, namely follicular development and the formation of the corpus luteum [9,10]. The growth of endothelial cells is regulated through a balance between endogenous proangiogenic factors, such as VEGF, and antiangiogenic factors, such as angiostatin and endostatin [8,11,12].

L-Mesitran Soft^®^ exhibits an angiogenic stimulation capacity comparable to VEGF; it is associated with higher levels of endothelial density and an apparent lower cellular toxicity [13]. This product includes in its composition 40% gamma-sterilized medical grade honey (MGH) and has been used over the past decades to treat wounds, given its antioxidant and antibacterial properties, as well as its ability to stimulate autolytic debridement, angiogenesis, cell migration and proliferation [14,15]. More recently, it has been studied as a natural cryoprotectant due to its antioxidant properties, which protect cells from thermal damage [16].

Vitamin D continues to stand out, with biological action far beyond the regulation of calcium metabolism and anticancer properties [17]. Vitamin D acts on follicular development during early folliculogenesis; however, the mechanisms involved and the necessary amounts of vitamin D are not fully understood [18]. Additionally, it seems to be able to influence angiogenesis, although its role is controversial; some studies suggest that it has antiangiogenic actions, while other reports claim proangiogenic effects [19,20,21,22]. 

Therefore, the aim of this study was to evaluate the application of a commercial product containing 40% MGH (L-Mesitran Soft^®^) in improving ovarian tissue endothelial density after transplantation and comparing it with VEGF and vitamin D supplementation.

## 2. Materials and Methods

### 2.1. Ethical Statement

The present study was approved by the Ethics Committee for Animal Experimentation (authorization number 11060495/23-11-2016) of the Faculty of Medicine of the University of Coimbra (FMUC) and performed according to European Guidelines and Portuguese Law. Animals were cared for according to the recommendations presented in the Guide for the Care and Use of Laboratory Animals of the National Institutes of Health. Experimental protocols were performed and reported according to ARRIVE guidelines [23].

### 2.2. Study Design

Given the ethical and practical restrictions of research in the field of human reproduction, animal models that show considerable similarities are often used. As such, the bovine model was used, given its physiological analogies (single ovulation, oocyte maturation and embryonic metabolism) and due to its similarity in size and structure. The rat model is more appropriate for tumor studies and postimplantation studies [24,25,26].

Bovine ovaries from 3 cows aged 14 to 18 months were collected at a regional slaughterhouse and transported to the laboratory within 1 h. Ovaries were dissected and the obtained ovarian cortical fragments were vitrified.

Afterward, warmed tissue samples were distributed into the following five groups of in vitro culture during a 48 h period: (1) control (absence of factor), (2) L-Mesitran Soft^®^ in vitro (0.2%, [13]), (3) L-Mesitran in vivo (control during culture and with direct application of L-Mesitran in the act of xenotransplantation), (4) VEGF (50 ng/mL [18]) and (5) vitamin D (100 Nm [15,19]. A sixth group (6) of fragments was thawed on the day of the transplant, without in vitro culture (direct transfer).

Of the 40 fragments of cryopreserved bovine ovarian tissue, 3 were analyzed immediately after thawing (group 0) and 14 samples were analyzed after 48 h of culturing. The remaining 23 fragments proceeded to the in vivo study; that is, to xenotransplantation into immunosuppressed rats. Seventeen grafts were removed after 7 days and the remaining six grafts, at 28 days after transplantation. The study design is represented in Figure 1.

Two sections for each probe and animal were evaluated using histological and immunohistochemistry techniques with antibodies anti-Ki-67, Caspase-3 and factor VIII.

### 2.3. Ovaries Collection, Vitrification, Thawing and Culture

In the laboratory, each ovary was divided into 2 identical parts, under laminar flow conditions, in a culture dish containing the same enriched solution (TCM-199 culture medium with 10% bovine serum). The medullary zone was removed with the aid of a scalpel blade, preventing any damage in the cortical tissue. After isolation, the cortical zone was cut into several fragments of approximately 1 cm^2^ and 0.5 mm of thickness.

Vitrification was performed, as follicular morphology is better preserved after in vitro culture [27]. Ovarian cortical fragments were first transferred to the equilibrium medium, containing 7.5% dimethyl sulphoxide (DMSO) and 7.5% ethylene glycol (EG) in TCM-199, supplemented with 20% newborn calf serum (NBCS), for 15 min at room temperature. They were subsequently transferred to the vitrification solution, containing 15% DMSO, 15% EG and 0.5 M sucrose in TCM-199 with 20% NBCS, for 2 min, also at room temperature. Ovarian pieces were placed in a 1.8 mL cryovial (Nunc CryoTubesT M, Catalog number: 363401) prefilled with liquid nitrogen, closed and then plunged into liquid nitrogen. Cryovials were stored in a tank until analysis.

For warming, cryovials were withdrawn from the liquid nitrogen tank using forceps, kept in air for 20 s and then placed in a water bath at 37 °C for 20 s. Thawed ovarian fragments were incubated in 1 mL of TCM-199 plus 20% NBCS solution supplemented with 1 M sucrose for 5 min, in the same cryovials. Ovarian pieces were then successively transferred to petri dishes containing TCM-199 plus 20% NBCS solution supplemented with 0.5 M, 0.1 M and 0.0 sucrose, in this order, respectively, for 5 min each.

Ovarian tissue fragments from groups 1 to 5 were cultured for 48 h with TCM-199 culture medium supplemented with 10% bovine serum. Culture medium was supplemented with L-Mesitran Soft^®^ in vitro at 0.2% for group 2, VEGF in a concentration of 50 ng/mL for group 4, and vitamin D in a concentration of 100 Nm for group 5. Cultivation was carried out at 37 °C in air with 5% CO_2_.

### 2.4. Experimental Animals

Twelve female Rowett nude rats (RNU, homozygous), aged 8–10 weeks, were provided by the animal facility of the FMUC. Rats were housed in individually ventilated cages under a 12 h light/dark cycle, with access to a standard diet and filtered water ad libitum. Paper rolls and strips were provided as environmental enrichment. Animals were distributed in a randomized manner throughout the study groups and identified by ear markings. Throughout the process, animal welfare was checked every 72 h; no significant issues were found throughout the study timeline.

### 2.5. Ovariectomy

All surgical procedures were performed under anaesthesia with sevofluorane (5%) and subcutaneous analgesia with carprofen (5 mg/kg, making a total of 0.2 mL per animal) [28,29,30].

For bilateral ovariectomy, each rat was placed in a supine position and the abdominal wall was shaved, cleaned and sterilized with povidon-iodine solution. A longitudinal median laparotomy was performed, with a 2 to 3 cm incision in the lower area of the abdomen (Figure 2). The ovaries were identified and removed after ligation of the vascular pedicle. After ensuring adequate hemostasis, the abdominal cavity was closed in layers [28,31]. Animals were kept without ovaries for 1 to 2 weeks, so that ovarian xenotransplantation would mimic clinical practice transplantation conditions in women with ovarian failure, and as per procedures in other xenotransplantation studies [5,32,33,34].

### 2.6. Xenotransplantation

After in vitro culture, tissue samples were randomly distributed among the animals. A single 2 cm incision was made in the abdominal wall, followed by blunt dissection of a 1 cm subcutaneous pocket using a pair of fine-curved watchmaker’s forceps (Figure 2). Fragments were inserted subcutaneously along the abdominal muscular fascia. Finally, the skin incision was closed using a single absorbable suture. Heterotopic transplantation is simpler, making the graft more accessible, and all locations are similar for short-time transplantations [34,35,36,37,38,39,40,41]. Thus, in this experimental work, and after the culture of hemiovaries, heterotopic autotransplantation of ovarian tissue was performed at the subcutaneous level. Grafts were recovered from each group on day 7 and 28 after grafting and the animals were immediately euthanized.

### 2.7. Histological Evaluation

Bovine ovarian tissue fragments were fixed in 4% buffered paraformaldehyde (VWR, Leuven, Belgium), included in a paraffin block and then serially sectioned at 3 μm thick in a microtome. The sections were stained with hematoxylin–eosin and then used for immunostaining.

Morphological analyses of ovarian tissue were performed via histological evaluation with the support of hematoxylin and eosin staining, according to the protocol performed by the Department of Pathology of Centro Hospitalar e Universitário de Coimbra. Slides were submitted to the following series of reagents: xylol (VWR, Fontenay-sous-Bois, France); absolute and 95% alcohol for 4 min; water for 2 min; Hematoxylin Gill 1 (Leica, Richmond, VA, USA); water to wash, eosin 1% aqueous solution (Bio-optica, Milan, Italy); 95% *v*/*v* alcohol for 30 s; absolute alcohol; and xylol (VWR, Fontenay-sous-Bois, France).

Observation of the slides was carried out via transmission microscopy (Axio Imager Z2 microscope) with ×20 magnification, a light intensity of 3.81 V, 0.8 DIC, a brightness of 23% and photographed with the Zen2 blue edition program (Carl Zeiss Microscopy GmbH, Jena, Germany, 2011) (Figure 3). Next, follicular classification was performed blindly by two independent observers using the Zen2 blue edition program (Carl Zeiss Microscopy GmbH, 2011). 

All follicles were counted and classified according to Gougeon classification (1986) as primordial (composed of a single layer of flattened granulosa cells), primary (presenting a single layer of cuboid granulosa cells) or secondary (composed by two or more layers of granulosa cells around the oocyte) [42]. Preantral and antral follicles were grouped with secondary follicles.

### 2.8. Immunohistochemistry Evaluation

Endothelial cell density was evidenced by immunostaining with anti-FVIII antibody (Cell Marque 760-2642, Bio-Rad Laboratories, Hercules, CA, USA), which is validated for microvessel staining [43] (Figure 4). Factor VIII is a glycoprotein exclusively synthesized by endothelial cells, specifically in Weibel-Palade bodies, and megakaryocytes. It usually binds to the von Willebrand factor, regulating the adhesion of thrombocytes to subendothelial connective tissue [44]. To evaluate cell proliferation and apoptosis, staining with anti-Ki-67 (PA5-19462, Thermofisher, Waltham, MA, USA; Figure 4) and anti-Caspase-3 (AHP2286, Bio-Rad Laboratories, Hercules, CA, USA; Figure 4) antibodies was performed, respectively. Briefly, blade dewaxing was carried out by heating to 72 °C, followed by cell conditioning (cc) and ultra CC, for 3 cycles. Next, preprimary peroxidase inhibitor was added to the primary antibody by heating the blade, followed by the application of a drop of anti-Caspase-3 antibody (1/100), anti-Ki-67 antibody (1/300) or anti-FVIII antibody (1/300) and incubation for 36 min. After this, color optimization reaction occurred using the Optiview system (OptiView DAB IHC Detection Kit, Ventana Medical Systems, Tucson, AZ, USA) and incubation with peroxidase HRP Multimer for 16 min. Finally, a drop of haematoxylin and a drop of bluing reagent (postcontrast) were applied with an incubation period of 4 min.

Slides were analyzed in the Axio Imager Z2 microscope by activating the ApoTome 2 and DIC-TLm imaging system, which is a microscope with thermal contrast lenses using differential interference. Images were acquired with a ×20 magnification and photographed with the aid of the Zen2 blue edition program (Carl Zeiss Microscopy GmbH, 2011), using 0.8 DIC, 4.0 V light and a brightness of 23% as constant settings.

Quantification of the stromal area marked by each antibody was carried out using Image J software (Fiji version, 1.8.0, USA) by two different evaluators. Five positions for each sample were randomly selected through the application of a rectangular grid (494 × 320 pixels), and the threshold was adjusted to the real image with the results subsequently presented as an average [45,46].

In addition, primordial, primary and secondary follicles were classified as positive or negative for Ki-67 and Caspase-3; positive classification was noted when staining was observed in the oocyte and at least half of the granulosa cells.

### 2.9. Statistical Analyses

Statistical analyses were performed using Statistical Analysis System (SAS) software 9.3 (SAS Institute Inc., Cary, NC, USA). PROC MIXED and PROC GLIMMIX were used to determine significant differences between groups and over time. The means of each treatment were calculated and compared using the PDIFF test of multiple comparisons. A *p*-value of less than 0.05 was considered statistically significant.

Throughout the text, significant differences obtained between groups and over time will be described and represented by different letters.

## 3. Results

### 3.1. Histological Evaluation

The number of primordial follicles was significantly higher in the L-Mesitran group in vivo 7 days after transplantation than in the other groups (*p* < 0.01) (Figure 5A,D). Regarding primary follicles, higher numbers were observed in fragments treated in vivo with L-Mesitran, 28 days after transplantation (*p* < 0.03) (Figure 5B,E). No significant differences were obtained for the number of secondary follicles (Figure 5C,F).

### 3.2. Endothelial Density

A higher endothelial density was observed after culture (48 h) in the presence of L-Mesitran (*p* = 0.0096), VEGF (*p* = 0.027) and vitamin D (*p* = 0.0028) (Figure 6B). These increments were not maintained 7 days after transplantation (Figure 6B). At day 28, the endothelial density of fragments cultured in the presence of L-Mesitran is significantly higher (*p* < 0.001) (Figure 6B).

Over time, the L-Mesitran in vitro group was the only group to present a higher percentage of positive cells for FVIII when compared to the tissue immediately after thawing (0 h) and over the three analyzed periods (*p* < 0.001). Additionally, within the 7-day groups, only the direct transfer group did not show higher values in relation to the tissue immediately after thawing (Figure 6C).

Moreover, the L-Mesitran in vivo group showed a significant increase in FVIII staining at day 7 after transplantation when compared to the in vitro culture of 48 h (*p* = 0.0028), which was not observed at other time points. In the case of vitamin D, a reduction in factor VIII expression/presence was observed between the mentioned time points (*p* = 0.0089) (Figure 6C).

### 3.3. Cell Proliferation

The fragment culture per se did not induce differences in cell proliferation between treatments (48 h). Moreover, 7 days after xenotransplantation, the fragments where L-Mesitran was administered in vivo were the only ones showing higher Ki-67 labelling than the control (*p* = 0.011). This higher proliferation was also significant in this group 28 days after xenotransplantation (*p* < 0.02, Figure 7D).

Follicular cell proliferation was significantly higher in all groups (*p* < 0.05) between 48 h and 7 days, except for the control group. In the case of the L-Mesitran in vivo group, this increase was extended to 28 days (*p* < 0.001), showing a more consistent increase in cell proliferation labelling over time (Figure 7C). The direct transfer group did not show a significant increase from 0 h to 7 days (Figure 7C).

Stromal labelling with anti-Ki-67 was also evaluated (Figure 7E). At 48 h, the VEGF group had a lower value than the control and the L-Mesitran in vitro and in vivo groups (*p* ˂ 0.003). The vitamin D group was also inferior to the control group (*p* = 0.013). At day 7, Ki-67 marking was higher in all groups in relation to the control (*p* < 0.05), except for the treatment of VEGF; the L-Mesitran in vivo group was superior compared to VEGF (*p* = 0.048) (Figure 7G). Finally, at day 28, a higher cell proliferation marking was evident in the L-Mesitran in vitro and VEGF groups compared to the control group (*p* = 0.0026 and *p* = 0.0077, respectively) (Figure 7G).

An initial increase in anti-Ki-67 immunostaining between 0 and 48 h (*p* < 0.008) was observed (Figure 7F) in all groups, except for the VEGF group. Furthermore, the L-Mesitran in vivo, vitamin D and direct transfer groups showed values higher at day 7 compared to 0 h (*p* < 0.03). Between 48 h and 7 days, the L-Mesitran in vivo, vitamin D and VEGF groups were the only ones in which no decrease in staining was observed, whereas in the case of VEGF, these values were not different from the control group (Figure 7F). Between days 7 and 28 there was a stabilization of the stromal immunostaining, with a decrease in only the L-Mesitran in vivo group (*p* = 0.018, Figure 7F).

### 3.4. Cell Apoptosis

No significant differences in follicular cell apoptosis among groups were obtained (Figure 8C).

Conversely, stromal labelling by anti-Caspase-3 antibodies show several significant differences between groups and over time (Figure 8E–G). At 48 h, the vitamin D group had a higher percentage of stroma marked with Caspase-3 (*p* < 0.01), except in comparison with the L-Mesitran in vitro group (Figure 8.G). At day 7 after transplantation, the L-Mesitran in vivo group displayed higher values compared to other groups (*p* < 0.02), except for the L-Mesitran in vitro group. Similarly, at 28 days, the L-Mesitran in vivo group showed the highest percentage of stromal labelling (*p* < 0.001) (Figure 8G).

As shown in Figure 8F, the vitamin D group was the only one in which Caspase-3 labelling increased from 0 to 48 h (*p* = 0.0055). In contrast, the L-Mesitran in vivo group was the only one in which there was no decrease in marking between 7 and 28 days (*p* < 0.001).

## 4. Discussion

Ovarian tissue cryopreservation with subsequent transplantation is the most promising technique for female fertility preservation, as it can restore both fertility and endocrine function, decreasing the consequences of premature ovarian failure in young cancer survivor patients. Furthermore, it is the only method to preserve fertility for prepubertal girls. Very recently, the American Society for Reproductive Medicine concluded that cryopreservation of ovarian tissue is no longer an experimental technique [3]. Therefore, it should be considered an established medical procedure with limited effectiveness that should be offered to carefully selected patients. In terms of efficacy, it can be compromised by follicular depletion that occurs after ovarian tissue transplantation, until tissue revascularization occurs [5]. This work analyzed several parameters, aiming to contribute to the improvement of this fertility preservation strategy.

Neo-angiogenesis is a key requirement to increase the survival of ovarian tissue after transplantation. Takae and Suzuki [47] reviewed therapeutic options that proved effective in this field, namely, growth factors such as VEGF, hormones and antioxidant and cell therapies. MGH is a therapeutic approach used for wound healing that also has angiogenic proprieties [48,49]; however, it had never been applied to ovarian tissue prior to the present study.

One of the critical factors for ovarian tissue transplantation is the ability to minimize ischemic injuries throughout the angiogenic process. Our results highlight the importance of in vitro culture, as the culture of the vitrified, warmed ovarian tissue prior to transplantation allowed establishment of conditions required for vascular development. When control and direct transfer groups were compared at day 7, the tissue that was previously under culture (control) had a higher endothelial density, with no differences in the follicular analysis.

Through histological and immunohistochemical evaluation, it was possible to analyze vascular and follicular dynamics in ovarian tissue grafts exposed to different treatments. According to Yang et al. (2008), the angiogenic process needs more than 48 h to be completed. During its development, a tissue graft is subjected to hypoxia situations responsible for follicular loss. Follicular survival and development depend in part on vascular growth, which, in turn, is dependent on the ability to be stimulated by angiogenic factors [39].

Through endothelial density evaluation, we demonstrated a progressive development of the vascular endothelium in almost all groups, essentially up to 7 days after xenotransplantation. However, the capacity for angiogenic stimulation of L-Mesitran in vitro up to 28 days after xenotransplantation stands out; it has a higher endothelial density compared to the other treatment options, demonstrating its importance as an alternative approach to using VEGF. Additionally, when we analyzed the effect of L-Mesitran in vitro treatment over time, after 48 h there was a significant increase in culture that remained after transplantation. This is in contrast with the other groups, in which endothelial density decreased after a longer transplantation period.

Taken together, these results and the higher stromal cell proliferation in the L-Mesitran in vitro group at day 28 after transplantation are in accordance with what has been described in [50], given that the endothelial cells present quite high mitotic rates until they reach the mature phase, providing adequate vascularization [51].

In the context of cell proliferation, L-Mesitran treatments (in vitro and in vivo) showed high levels of Ki-67 expression in stromal cells 7 days after transplantation. It was possible to demonstrate that cell recovery occurs over time when compared to control. These findings are in agreement with the results of Gastal et al., who reached proliferative potential and tissue survivability after 7 days in culture [52]. Furthermore, several authors [28,31,50] describe the presence of necrotic regions in the first days after transplantation. 

Regarding cell death, there were significant differences in stromal cells’ apoptosis in the present study. The group with the highest expression was L-Mesitran in vivo, which may be correlated with the high cellular activity observed through Ki-67 staining. This may be related to the cell turnover mechanism, i.e., the remodeling process necessary for cell differentiation and enucleation, which closely correlate with Caspase-3 activation. Although caspases are related to cell apoptosis, caspase-mediated proteolysis also controls several non-lethal cellular activities, such as proliferation and cytoskeleton reorganization, among others [53]. Thus, a greater endothelial density and cell proliferation, mediated by higher apoptosis, seems to be related to normal processes in terms of the adaptation and remodeling of an ovarian cortex graft [43].

Historically, ovarian research has focused on its functional units, the follicles. The ovarian stroma is gaining in importance, as it contains several cells and components that are crucial to complex ovarian dynamics [54]. Therefore, the viability of the stroma was also analyzed. With the application of MGH, high rates of vascularization were observed, essentially up to 7 days after transplantation, in parallel with better tissue viability as evaluated by staining with Ki-67 and Caspase-3.

Follicle viability, as previously mentioned, can be threatened by hypoxia and the consequent ischemic injury that occurs after transplantation; therefore, the expression of Ki-67 and Caspase-3 was assessed in granulosa cells. In the present study, there were no significant differences in granulosa cells’ apoptosis. Regarding cell proliferation, only the L-Mesitran in vivo group evidenced the ability to stimulate follicular development. The same effect was observed with the administration of antiapoptotic agents before freezing and by injection after transplantation [55].

However, according to Winkler-Crepaz et al. (2008), after the transplantation of ovarian tissue cortex, follicular loss is unlikely to occur solely due to ischemic apoptosis. Therefore, the high recruitment of primordial follicles is a mechanism to consider. In this way, the density and viability of follicles present in the cortex of the transplanted ovarian tissue is significant [56]. The L-Mesitran in vivo group may have higher follicular recruitment, as it had a higher number of primordial follicles at day 7 and primary follicles at day 28 after transplantation, in parallel with higher levels of follicular proliferation and apoptosis.

The major strength of this study is that it is the first to evaluate and compare a product containing medical grade honey with other strategies to improve ovarian tissue transplantation. However, some points, such as an increase in sample size and the evaluation of endocrine function, can addressed in future research.

## 5. Conclusions

L-Mesitran Soft^®^ seems to have a positive influence on ovarian tissue xenotransplantation. It demonstrated potential for revascularization with apparent low cytotoxicity. Additionally, the results obtained regarding cell proliferation are very promising and seem to be in line with the apoptosis rates that were detected and necessary, from the perspective of cell turnover, for tissue remodeling after ovarian transplantation. It must also be stressed that L-Mesitran Soft^®^ is an inexpensive product and easy to handle, which are also advantages over other potential treatments.

The animal model used, which involved xenotransplantation of bovine ovarian cortex in ovariectomized immunodeficient rats at the level of abdominal muscular fascia, was consolidated as a model applicable to other studies and products in the field of human fertility.

Finally, we can conclude that the incorporation of L-Mesitran Soft^®^ may constitute an additional treatment to improve transplanted ovarian tissue vascularization and viability, either through its administration in vitro (culture medium) or in vivo (transplant site); thus, improving graft survival and transplant durability in women’s fertility preservation.

## Figures and Tables

**Figure 1 bioengineering-09-00357-f001:**
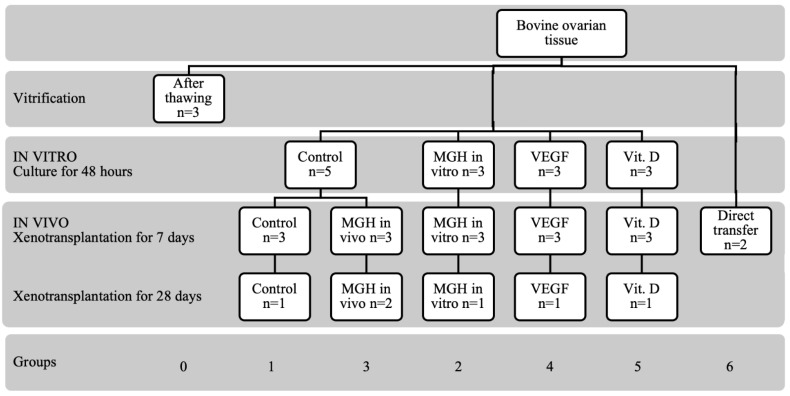
Experimental layout. Cortical fragments of bovine ovarian tissue were cryopreserved, thawed and distributed to each group. Forty ovarian fragments were removed for histological and immunohistochemical analysis at each stage of the study. The number (*n*) indicates the number of different bovine samples in each condition; all were analyzed in two sections by two independent investigators. MGH, medical grade honey—L-Mesitran Soft^®^ 0.2%; VEGF, vascular endothelial growth factor; Vit.D, vitamin D.

**Figure 2 bioengineering-09-00357-f002:**
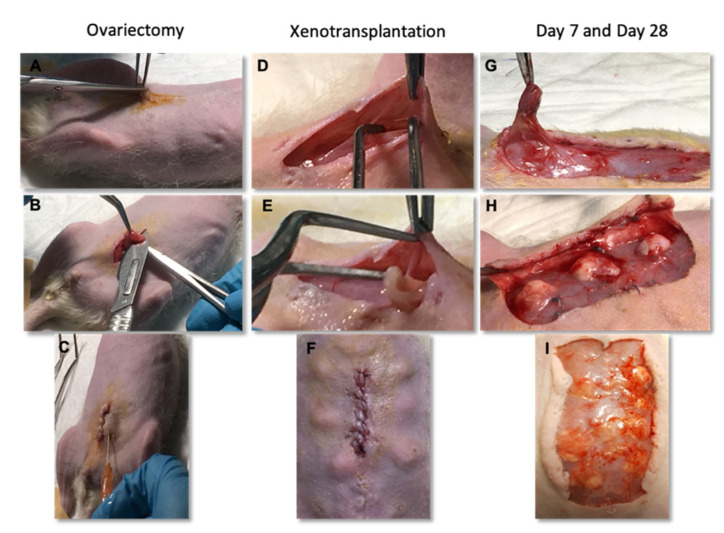
Surgical procedures performed on female Rowett nude rats. For ovariectomy, a longitudinal median laparotomy in the lower area of the abdomen was performed (**A**), the ovaries were identified and removed (**B**) and, after abdominal wall suture, they were administrated a subcutaneous analgesia (**C**). Bovine ovarian tissue was transplanted after a blunt dissection of a subcutaneous pocket (**D**–**F**). Fragments were recovered after 7 (**G**,**H**) and 28 days (**I**).

**Figure 3 bioengineering-09-00357-f003:**
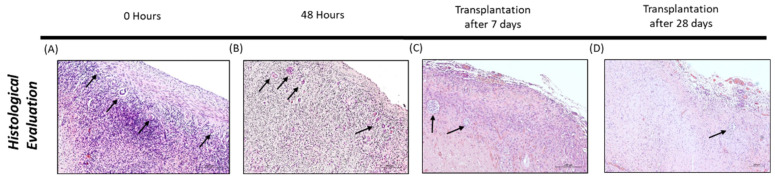
Histological evaluation of bovine ovarian fragments immediately after thawing (**A**), from the control group of in vitro culture (**B**) and after transplantation (**C**,**D**). Arrows indicate some of the identified follicles in the tissue. Source: authors’ images.

**Figure 4 bioengineering-09-00357-f004:**
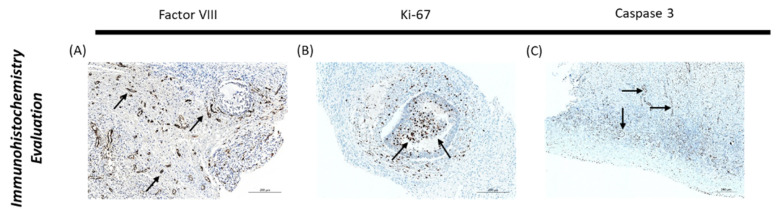
Immunohistochemistry evaluation of ovarian tissue. (**A**) Immunostaining with factor VIII. (**B**) Immunostaining with Ki-67. (**C**) Immunostaining with Caspase-3. Arrows indicate positive staining. Source: authors’ images.

**Figure 5 bioengineering-09-00357-f005:**
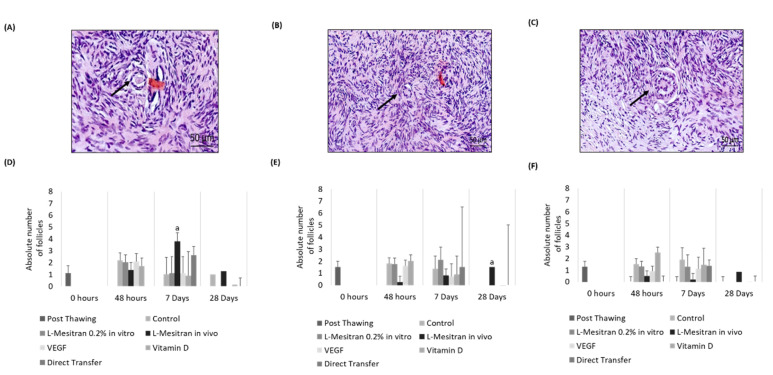
Follicular Quantification. (**A**,**D**) Primordial Follicles. (**B**,**E**) Primary Follicles. (**C**,**F**) Secondary Follicles. Arrows indicate some of the identified follicles in the tissue; a: indicate statistical differences (*p* ≤ 0.05). Source: authors’ images.

**Figure 6 bioengineering-09-00357-f006:**
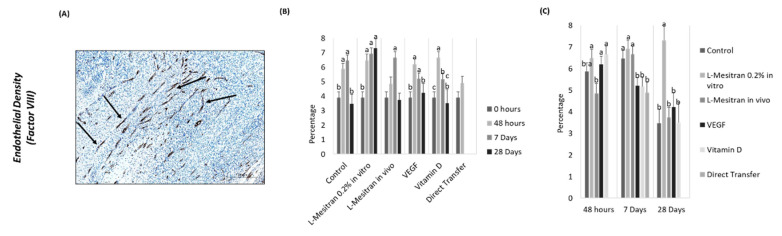
Endothelial Density in ovarian tissue. (**A**) Immunostaining with factor VIII (arrows indicate positive staining). (**B**) Endothelial density, with factor VIII marking at 48 h, 7 and 28 days. (**C**) Endothelial density over time, with factor VIII marking. Arrows indicate positive staining. a–c: indicate statistical differences (*p* ≤0.05). Source: authors’ images.

**Figure 7 bioengineering-09-00357-f007:**
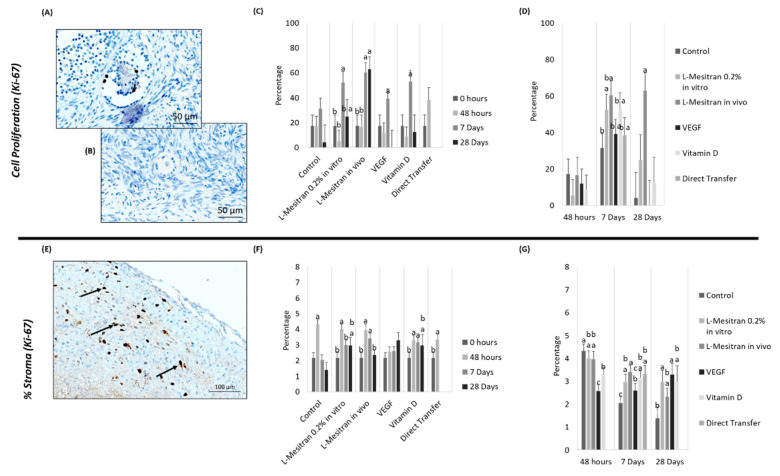
Cell Proliferation in ovarian tissue. (**A**) Follicle with positive marking for Ki-67. (**B**) Follicle with negative marking for Ki-67. (**C**) Cellular proliferation in the follicles, with Ki-67 marking at 48 h, 7 and 28 days. (**D**) Cellular proliferation in the follicles, with Ki-67 marking over time. (**E**) Stroma quality marking with Ki-67 (arrows indicate positive staining). (**F**) Stroma quality, with Ki-67 marking at 48 h, 7 and 28 days. (**G**) Stroma quality, with Ki-67 marking over time. Arrows indicate positive staining. a–c: indicate statistical differences (*p* ≤ 0.05). Source: authors’ images.

**Figure 8 bioengineering-09-00357-f008:**
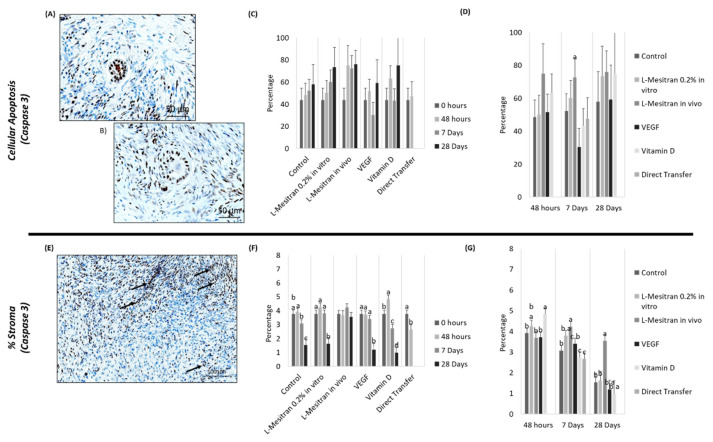
Cellular Apoptosis in ovarian tissue. (**A**) Follicle with positive marking for Caspase-3. (**B**) Follicle with negative marking for Caspase-3. (**C**) Cellular apoptosis in the follicles, with Caspase-3 marking at 48 h, 7 and 28 days. (**D**) Cellular apoptosis in the follicles, with Caspase-3 marking over time. (**E**) Stroma quality marking with Caspase-3 (arrows indicate positive staining). (**F**) Stroma quality with Caspase-3 marking at 48 h, 7 and 28 days. (**G**) Stroma quality with Caspase-3 marking over time. Arrows indicate positive staining. a–c: indicate statistical differences (*p* ≤ 0.05). Source: authors’ images.

## Data Availability

No new data were created or analyzed in this study. Data sharing is not applicable to this article.

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
