# Peer review of "Medical Grade Honey as a Promising Treatment to Improve Ovarian Tissue Transplantation"

_bioengineering, 2022, doi:10.3390/bioengineering9080357_

Round 1

Reviewer 1 Report

Line 108 states there are 5 groups, but elaborate 6. Will be helpful to write the number of replicates in each arm in figure 1. It seems that the number of replicated per condition is pretty low. Why ovariectomy and transplantation were performed separately? Why this site of implantation was selected? What is the rational for selecting these specific time points? Especially when looking for apoptosis is one of the goals. Figure 5, counting primordial follicles: the follicle in box a looks like a primary, counting absolute number of follicles is not a good way to compare when comparing slivers of an ovary. The primary follicle in B has poor morphology. How can you explain there were more primordial (5d) after a week, more primary after 28 days (5e), but no difference in secondaries. How come the endothelial density is so different between 2 days in vitro to 1 weeks in vivo? The turnover of blood vessels is not that quick, how do you explain the different effects, and according to the results is these controls are appropriate. Figure 7 the follicle positive to ki67 is surprising that so many cells are positive, most probably the follicle is not healthy, and will not survive. The other follicle, which is negative for the staining has poor morphology, so it is not surprising that a follicle at this developmental stage no staining was detected. I am also surprised that there's so much proliferation within the stroma. Apoptosis is relatively an acute event. The idea of detecting apoptosis related to an exposure that was done 4 weeks earlier is concerning, is there a deleterious continuous damage conferred by the Tx? High Ki67 levels are not equal a healthy tissue. Indeed, it is hard to assess viability, bur survival of normal morphologically follicles can assist in that evaluation. But even morphologically the distance between nuclei of GC, lack of the normal halo of TC. Theses follicles are dying, or already dead. There is normal turnover of proliferation and apoptosis in large antrals growing, this is not what we see here.

Author Response

Reviewer: can and must be improved sections

Authors: Thank you very much for your suggestion, we add relevant and reviewed the references; with the reviewers’ suggestions, we improved the description of the research design, methods, results, and conclusions.

Reviewer: Line 108 states there are 5 groups, but elaborate 6. Will be helpful to write the number of replicates in each arm in figure 1. It seems that the number of replicated per condition is pretty low.

Authors: Thank you very much for your suggestion, we add the number of cases in the experimental layout. The change in the manuscript was made in L.114-116 and in figures’ legend. We have a low number, but it was analysed in two sections of each ovarian fragment and by two independent investigators. We also add this limitation of the study in lines 439-442.

Reviewer: Why ovariectomy and transplantation were performed separately?

Authors: It is a pertinent question, we decided to perform ovariectomy and not in the same day of transplantation procedure, as we pretend to mimic the conditions of transplantation in clinical practice in women with ovarian failure. Another limitation in the ovarian tissue transplantation is the impact in follicular pool after the procedure, due to follicles burn-out. The follicles activation compromises the longevity of the transplant. So, with previous ovariectomy, the rats were in a “menopausal” status, and we can see the impact in the follicular pool. Actually, it has already been demonstrated that a significant increase of secondary or more mature follicles was observed in ovaries transplanted into ovariectomized mice accompanied with a decrease in primary follicle density compared to ovaries transplanted into non-ovariectomized mice. One of the likely explanations for the transplantation-induced activation of primordial follicles in ovariectomized mice lies in the absence of the mice own ovaries [1].

Additionally, this option of doing ovariectomy at least one week before xenotransplantation, has already been done in other studies with xenotransplantation.

The change in the manuscript was made in L.169-171.

Reviewer: Why this site of implantation was selected?

Authors: We choose subcutaneous transplantation site regarding a previous study that compared different graft sites [2], and showed that they are similar for the studies of the transplant up to 21 days [3]. Additionally, many studies in this field with xenotransplantation used the same site of implantation for studies from 28 days up to 6 weeks [4]–[9]. Finally, heterotopic transplantation is simpler, making the graft more accessible [10]. Ovarian tissue transplantation can be orthotopic, when it is reimplanted in the remaining ovary or in the peritoneum at the level of the ovarian pit, or heterotopic, when performed elsewhere, namely subcutaneously, intramuscularly or retroperitoneally. Orthotopic transplantation is more invasive, but in clinical practice it has been shown to be more effective.

The subcutaneous tissue of the abdomen is, in the mouse, one of the locations that provides the most adequate physiological environment for follicular development ([11][12]). Luyckx et al. (2013), demonstrated the existence of high follicular survival rates of cryopreserved prepubertal ovarian tissue from mice, through the retention of a large pool of dormant primordial follicles in the graft ([13]). In the human species, in 2013, the first clinical pregnancy after heterotopic ovarian tissue transplantation was reported, it has been performed in the anterior abdominal wall (in subperitoneal location), in a patient previously submitted to bilateral oophorectomy ([14]).

Thus, in this experimental work and after the culture of the hemi-ovaries, heterotopic ovarian tissue transplantation was performed at the subcutaneous level.

The change in the manuscript was made in L.186-188.

Reviewer: What is the rational for selecting these specific time points?

Authors: The transplant was performed for 28 days, considering the transplant site, the recovery of endocrine function and the folliculogenesis. Heterotopic transplantation at the subcutaneous level, in addition to being technically simpler and more accessible, is similar from the follicular point of view to other locations, considering the duration of transplantation up to 21 days [3]. Some studies use it until 28 to 30 days [5], [6], [9], [15], [16], more than that can compromised the tissue, as the environment is not as good as in orthotopic site transplantations. Additionally, folliculogenesis is a lengthy process and having 28 days it is possible to see some differentiation in follicles stage.

In addition to the evaluation of the graft at the end of the experiment, the tissue was also studied at 7 days, since reperfusion of the ovarian tissue at the end of that time has been demonstrated in rat studies [17]–[19]. The revascularization period is a crucial step for follicular survival from ischemic damage, as the implanted tissue fragments do not contain blood vessels. Although it is not possible to determine the longevity of the transplanted ovarian tissue, it is known that the graft needs 4 to 5 days to be reoxygenated. In the case of rats, functional vessels were detected from the 7th day and in humans from the 5th day after transplantation, followed by gradual oxygenation over the next 5 days ([17]).

Reviewer: Especially when looking for apoptosis is one of the goals. Figure 5, counting primordial follicles: the follicle in box a looks like a primary, counting absolute number of follicles is not a good way to compare when comparing slivers of an ovary. The primary follicle in B has poor morphology. How can you explain there were more primordial (5d) after a week, more primary after 28 days (5e), but no difference in secondaries.

Authors: In this study we observe more primordial (Figure 5d) after a week [20], more primary after 28 days (Figure 5e), but no difference in secondaries (Figure 5.f) due to the short study period. Th increase in primary follicles could be explained by the normal folliculogenesis. If we analyse the chronology of folliculogenesis (on figure below) in human ovaries the preantral period takes 300 days for a recruited primordial to grow and develop to an early antral stage. With in vitro culture there is a significant change of follicles from the quiescent phase to the growth pool during short periods of culture of 6 to 10 days, in the particular case of bovine ovarian tissue it is observed a repeated and extensive primordial activation over 48h in vitro [21], [22] indicating that the activation results from a release of intraovarian factors that act as inhibitors of follicular growth [21], this could be one of the factors that led to the detection of a significantly higher number of primordial and primary follicles. So, if the increase in primary follicles would be seen earlier it could be by the abnormal follicle activation after ovarian tissue transplantation. Follicular activation is one of the already described limitations of cryopreservation and ovarian tissue transplantation, that is responsible for follicular loss immediately after cryopreserved ovarian tissue transplantation, which limits its success and longevity [23].

Figure from: Gougeon, A. (1996). Regulation of ovarian follicular development in primates: Facts and hypotheses. Endocrine Reviews, 17(2), 121–155. https://doi.org/10.1210/edrv-17-2-121

Reviewer: How come the endothelial density is so different between 2 days in vitro to 1 weeks in vivo? The turnover of blood vessels is not that quick, how do you explain the different effects, and according to the results is these controls are appropriate.

Authors: If we observe the results in Figure 6, we can see that the percentage mean endothelial density at 48 hours and 7 days changes between 5% and 7% at both times.

Endothelial density, assessed by Factor VIII, demonstrated a progressive development of the vascular endothelium, essentially until 7 days after xenotransplantation [17]–[19]. These results coincide with what is described in the literature [24][25] as endothelial cells show very high mitotic rates until they reach the mature stage, providing adequate vascularization [25].

The assessment of the tissue after culture and before transplantation, allows us to understand de effect of the culture and the impact of the transplant. Some studies also use an early time-point at day 2 to 3, but after xenotranplantation [4], [26]–[28]

Regarding the angiogenesis from a chronological point of view, Bentley and Chakravartula described that it requires multiple cycles of gene regulation, which alter protein synthesis and last about 4 to 6 hours [29]. The change in the position of endothelial cells during sprouting occurs, on average, every 3.7 hours, and allows for the branching of new vessels [29].

Reviewer: Figure 7 the follicle positive to ki67 is surprising that so many cells are positive, most probably the follicle is not healthy, and will not survive. The other follicle, which is negative for the staining has poor morphology, so it is not surprising that a follicle at this developmental stage no staining was detected. I am also surprised that there's so much proliferation within the stroma.

Authors: thank you very much for your suggestion. The change on Figure 7 was made in L.327-328.

Regarding the comment about the levels of proliferation in stroma, we understand you, as when the percentage of ki-67 positive cells is higher than the physiological one, it is considered that we are facing a tumoral situation [30], [31]. Proliferation rate in stroma the groups of study is between 1 to 6% as it is represented in figure 7. That is not too much compared to th tissue analysed immediately after vitrification (2%), also a previous study of our group that analysed proliferation rate in stroma of rat ovarian in culture [32] and Vatanparast et al. [33] that used sheep ovarian tissue. Actually, there is insufficient data in the physiological levels of proliferation in ovarian stroma as it is frequently not analysed in the studies of this field. For many years, ovarian research has focused on folliculogenesis. Recently, the ovarian stroma has become a new focus of investigation, critical for understanding the complex dynamics of the ovary [34].

Reviewer: Apoptosis is relatively an acute event. The idea of detecting apoptosis related to an exposure that was done 4 weeks earlier is concerning, is there a deleterious continuous damage conferred by the Tx?

Authors: Cellular apoptosis is one of the essential mechanisms in homeostasis, pathophysiology and normal development of the ovaries [35], [36].

Within the framework of a cell turnover mechanism, that is, the cell remodelling processes necessary for differentiation and enucleation, which are closely correlated with the activation of Caspase 3. Although Caspases are related to cell apoptosis, proteolysis mediated by caspases also controls several non-lethal cellular activities, such as proliferation, cytoskeletal reorganization, among others. In fact, in embryonic stem cells, an increase in the activity of Caspases enables the activation of transcription factors that are central to differentiation and, therefore, to the regulation of pluripotency [37]. Thus, greater endothelial density and general cell proliferation, accompanied by superior apoptosis, seems to be related to normal processes in terms of adaptation and remodelling of an ovarian cortex graft.

Regarding the quality of the stroma, evaluated through caspase 3, the results observed are in line with the levels of cellular apoptosis observed in the follicles at 48 hours and 7 days. In particular, in addition to the in vivo L-Mesitran group, treatment with Vitamin D in culture (48h) shows significantly elevated levels compared to 7 and 28 days after xenotransplantation, thus indicating tissue recovery over time. According to Winkler-Crepaz et al. (2008), after transplantation of ovarian tissue cortex it is unlikely that follicular loss will occur solely due to ischemic apoptosis, and the high recruitment of primordial follicles is a mechanism to consider. In this way, the density and viability of the follicles present in the cortex of the transplanted ovarian tissue is fundamental, since the grafts must have an intermediate density, in order to prevent follicle depletion, resulting from the intrinsic pressure after transplantation, and thus allowing a long duration of normal ovarian function [38], [39]. Additionally, the presence of stromal fibrosis may be positively associated with follicular activation, since stromal rigidity appears to contribute to follicular dynamics [40], thus explaining the high levels of cell proliferation and apoptosis observed.

Reviewer: High Ki67 levels are not equal a healthy tissue. Indeed, it is hard to assess viability, bur survival of normal morphologically follicles can assist in that evaluation. But even morphologically the distance between nuclei of GC, lack of the normal halo of TC. Theses follicles are dying, or already dead. There is normal turnover of proliferation and apoptosis in large antrals growing, this is not what we see here.

Authors: It is a valid comment, the follicular degeneration is observed in culture and after transplantation. The great potential of this study is that it is the first that evaluates and compares the application of a product containing 40% medical grade honey with other strategies to improve ovarian grafts viability.

Reviewer 2 Report

This manuscript evaluated and compared the application of L-Mesitran Soft®, a product containing 40% medical grade honey (MGH), with other strategies to improve ovarian grafts viability. The work is novelty, however it is not rigorous, for example, the following points need to be addressed.

1.      There was a confusion in the group setting. To demonstrate the efficacy of L-mesitran xenotransplantation, comparisons may be made with control xenotransplantation will better.

2.      By reading the manuscript, the reader does not know why the concentration of MGH, VEGF and Vit. D was chosen?

3.      Figure should list the representative pictures of each group, not only positive group will be better.
